# ITERATIVE VECTORS: BOOST IN-CONTEXT LEARNING WITHIN ACTIVATIONS

## ABSTRACT

In-context learning (ICL) has emerged as a standard paradigm for utilizing language models. Although ICL is convenient due to the absence of backpropagation, selecting and processing appropriate demonstration examples can be difficult and time-consuming, particularly when the number of examples is large. We propose to explore the potential of activation space through Iterative Vectors (IVs), a technique designed to enhance in-context performance and necessitating only forward inference passes. IVs are employed by first extracting and iteratively steering activations within a language model, then applying them during inference with minimal computational and memory overhead. We evaluate IVs across numerous tasks using four popular models and observe significant improvements. Our findings suggest that activation steering can serve as a promising direction for in-context learning, thereby opening new avenues for future research.

## 1 INTRODUCTION

Few-shot learning has long been a prominent research focus. Recently, language models (LMs) have shown the capability to execute few-shot learning through in-context learning (ICL) (Brown et al., 2020). In this approach, learning a new task involves conditioning on a few support examples and predicting the most suitable tokens to complete a query input, all without the need for any parameter updates. This method is appealing because it relies solely on inference, allowing for quick adaptation to various downstream tasks.

However, it has been noted that despite its potential, the predictions of LMs can be highly volatile when conditioned on prompts. The outcomes depend significantly on the templates, demonstrations, their permutations, and can even ignore or violate the instructions of the prompt (Webson & Pavlick, 2022; Min et al., 2022b). This finding is also corroborated in our experiments, wherein adding more in-context examples does not always result in improvements. Instead, it introduces uncertainty, which compromises LMs' reliability and usability. Furthermore, in theory, the inference

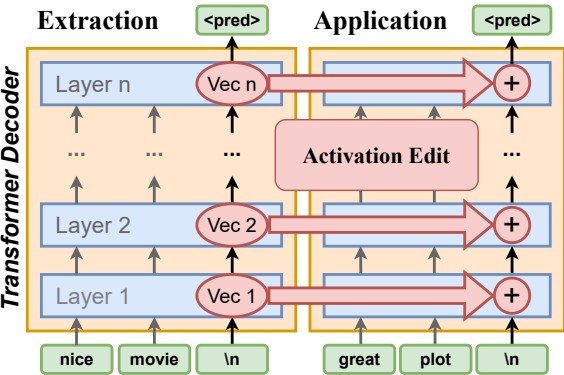

Figure 1: A general illustration of how activation vectors improve ICL performance by extracting and editing model activations.

time increases quadratically as more examples are appended to the query. When the examples are lengthy, it may be unfeasible to accommodate them within the desired timeframe and the model's context length.

In this paper, we introduce Iterative Vectors (IVs) to offer a new perspective. As illustrated in Figure 1, rather than staying in the discrete prompt space, IVs delve into the extensive activation space of the model. This exploration reveals a largely uncharted area for developing new methods, with our pioneering efforts to demonstrate how ICL can be enhanced from the representations within the model.

IVs are generated by extracting the difference of attention activations from queries with and without preceding examples during the inference process, with the goal of capturing the insights the model learns from the input examples. These IVs are then iteratively reintroduced into the model, facilitating the formation of more stable and effective vectors while continuously incorporating information from subsequent examples. Subsequently, these IVs can be utilized in future inference procedures. This methodology does not impede the ICL framework and incurs minimal computational and memory overhead, thereby making our method more advantageous to use.

IVs can substantially enhance ICL performance. When evaluated across 4 models and 13 diverse tasks, IVs outperformed standard ICL baselines by an average margin of 3.5%, and also exceeded the performance of two other activation vector methods (Section 4). Furthermore, IVs demonstrate significant time savings in achieving boosted one-shot performance (Section 4.1). They also effectively scale with the quantity of demonstration shots preceding the query (Section 4.2). Whether supplied with only a few or numerous examples for extraction, IVs consistently adapt to the given task, maintaining a trajectory of improved performance (Section 4.3). Finally, through ablating the hyperparameters of our method, we discovered an optimal interaction among them that maximizes performance, thereby affirming that each is an essential component of the methodology (Section 4.4).

Our contributions are highlighted as follows:

1. We establish the evaluation framework for activation vectors in the ICL setting and adapt two preliminary activation vector methods to this framework.

2. We propose a novel activation vector method specifically designed for ICL, termed Iterative Vectors (IVs), which enhances ICL performance without the need for backpropagation.

3. Extensive experiments demonstrate that our method exhibits superior performance and underscores the potential of activation vectors for ICL.

To the best of our knowledge, we are the first to investigate the application of activation vectors on diverse real-world in-context learning tasks and to demonstrate their potential with in-context examples during inference.

## 2  RELATED WORK

Some preliminary studies have investigated the manipulation of language models within the representation space by utilizing lightweight vectors, which we refer to as *activation steering* with *activation vectors* in this paper.

Activation steering methods contrasts with existing prompt tuning methods (Li & Liang, 2021; Lester et al., 2021), which operates in a continuous parameter space but still as part of the prompt and requires training via backpropagation. Again, unlike Parameter-Efficient Fine-Tuning (PEFT) methods, e.g. LoRA (Hu et al., 2021), they does not seek to tune the parameters of the model but rather modifies the activations during inference.

### 2.1  ACTIVATION VECTORS

Task Vectors (Hendel et al., 2023) are extracted from one layer of the model during ICL inference and then applied to a zero-shot query to determine whether they can preserve task-relevant information. Function Vectors (Todd et al., 2023), on the other hand, select activations from the top attention

heads, based on their causal effect in generating the correct response. These selected activations are then averaged and introduced into a specific layer of the model.

Although these two methods align closely with our approach and share similar objectives, their primary testing has been limited to straightforward synthetic tasks, such as identifying antonyms, naming country capitals, and providing plural forms, rather than ICL tasks with demonstrations. Consequently, the practical applicability of these vectors in real-world environments remains uncertain.

In contrast, our objective is to conduct evaluations within a more realistic context by utilizing real-world classification datasets. This approach aims to offer a more thorough assessment framework for activation vectors. We have adapted and included these two methods for comparison to facilitate the practical application of activation vectors beyond theoretical constructs.

## 2.2 GENERATIVE STEERING

Another research direction focuses on modifying LMs' activations for generation and transfer purposes. Latent Steering Vectors (Subramani et al., 2022) aim at sentence recovery and sentiment transfer. Inference-Time Intervention (Li et al., 2023) involves probing each attention head and guiding the model with the probe vector to enhance the truthfulness of the generated text. Studies by Turner et al. (2023) and Liu et al. (2024) address style and sentiment transfer by employing positive and negative sentence pairs to extract contrastive guidance.

Despite their shared similarities in operating within the representation space, these methods either necessitate training with backpropagation or are specifically tailored for generative or transfer tasks between sentence pairs. Consequently, it is not immediately clear how they should be integrated into the ICL setting, which we leave for future research.

## 3 METHOD

In this section, we begin by establishing the theoretical foundation of our method. Following this, we outline the evaluation protocols to clearly define the relevant notations. Finally, we present our method in detail.

## 3.1 THEORETICAL FOUNDATION

Given the significance of in-context learning, numerous theories have been proposed to explain its underlying mechanisms, as evidenced by Xie et al. (2022); Chan et al. (2022); Ye et al. (2023); Oswald et al. (2023). One particularly intriguing line of hypothesis posits that a pretrained LM operates as a meta-optimizer, generating meta-gradients which it then applies to address ICL tasks. We now present an overview of this concept.

First, let us revisit the dual form of the perceptron and apply it in the modern context of deep NNs (Irie et al., 2022). Formally, assume a linear layer trained via gradient descent utilizing $T$ training inputs $(\boldsymbol{x}_1, \ldots, \boldsymbol{x}_T)$ and their corresponding (backpropagated) error signals $(\boldsymbol{e}_1, \ldots, \boldsymbol{e}_T)$, where $\boldsymbol{x}_t \in \mathbb{R}^{d_{in}}$ and $\boldsymbol{e}_t \in \mathbb{R}^{d_{out}}$. If standard gradient descent is applied, a loss function $\mathcal{L}$ produces the error signal $\boldsymbol{e}_t = -\eta_t (\nabla_{\boldsymbol{y}} \mathcal{L})_t$, where $\eta_t \in \mathbb{R}$ is the learning rate, and $\boldsymbol{y}_t = \boldsymbol{W} \boldsymbol{x}_t$ is the output of the linear layer. Its weight matrix is given by

$$\boldsymbol{W} = \boldsymbol{W}_0 + \sum_{t=1}^{T} \boldsymbol{e}_t \otimes \boldsymbol{x}_t, \tag{1}$$

where $\boldsymbol{W}_0 \in \mathbb{R}^{d_{out} \times d_{in}}$ represents the initial value of the weights. This linear layer transforms an input $\boldsymbol{x} \in \mathbb{R}^{d_{in}}$ into an output $S_1(\boldsymbol{x}) \in \mathbb{R}^{d_{out}}$:

$$S_1(\boldsymbol{x}) = \boldsymbol{W} \boldsymbol{x}. \tag{2}$$

Next, consider a composite layer $S_2$ that stores $T$ key-value pairs, $(\boldsymbol{x}_1, \boldsymbol{e}_1), \ldots, (\boldsymbol{x}_T, \boldsymbol{e}_T)$, represented by a key matrix $\boldsymbol{X} = (\boldsymbol{x}_1, \ldots, \boldsymbol{x}_T) \in \mathbb{R}^{d_{in} \times T}$ and a value matrix $\boldsymbol{E} = (\boldsymbol{e}_1, \ldots, \boldsymbol{e}_T) \in$

$\mathbb{R}^{d_{out} \times T}$, along with a weight matrix $\boldsymbol{W}_0 \in \mathbb{R}^{d_{out} \times d_{in}}$. This layer transforms an input $\boldsymbol{x} \in \mathbb{R}^{d_{in}}$ into an output $S_2(\boldsymbol{x}) \in \mathbb{R}^{d_{out}}$ by

$$S_2(\boldsymbol{x}) = \boldsymbol{W}_0\boldsymbol{x} + \mathrm{Attn}(\boldsymbol{X}, \boldsymbol{E}, \boldsymbol{x}), \tag{3}$$

where the parameters of the unnormalized attention operator $\mathrm{Attn}(\cdot)$ are, in order, the key, value, and query.

It can be shown that $S_1$ and $S_2$ are equivalent by expanding the attention operation as

$$\mathrm{Attn}(\boldsymbol{X}, \boldsymbol{E}, \boldsymbol{x}) = \boldsymbol{E}\boldsymbol{X}^\top\boldsymbol{x} = \left(\sum_{t=1}^{T} \boldsymbol{e}_t \otimes \boldsymbol{x}_t\right)\boldsymbol{x}. \tag{4}$$

This expression elucidates that the forward operation of any linear layer in neural networks, trained via gradient descent, can be interpreted as a key-value-query attention mechanism (Vaswani et al., 2017). In this framework, the training data points act as the keys, the corresponding gradients serve as the values, and the test input generates the query.

Utilizing the dual form, ICL can be interpreted as a meta-optimization process (Dai et al., 2023). This was achieved by reversing the direction of the equivalence and breaking down the attention key and value terms for the ICL query token into its zero-shot and demonstration components, as formally expressed:

$$\widetilde{\mathcal{F}}_{\mathrm{ICL}}(\boldsymbol{q}) = \boldsymbol{W}_{\mathrm{ZSL}}\boldsymbol{q} + \mathrm{LinearAttn}\left(\boldsymbol{W}_V\boldsymbol{X}', \boldsymbol{W}_K\boldsymbol{X}', \boldsymbol{q}\right) \tag{5}$$

$$= \boldsymbol{W}_{\mathrm{ZSL}}\boldsymbol{q} + \sum_i \boldsymbol{W}_V\boldsymbol{x}'_i\left(\left(\boldsymbol{W}_K\boldsymbol{x}'_i\right)^T\boldsymbol{q}\right) \tag{6}$$

$$= \boldsymbol{W}_{\mathrm{ZSL}}\boldsymbol{q} + \sum_i \left(\left(\boldsymbol{W}_V\boldsymbol{x}'_i\right) \otimes \left(\boldsymbol{W}_K\boldsymbol{x}'_i\right)\right)\boldsymbol{q} \tag{7}$$

$$\triangleq \boldsymbol{W}_{\mathrm{ZSL}}\boldsymbol{q} + \Delta\boldsymbol{W}_{\mathrm{ICL}}\boldsymbol{q} \tag{8}$$

$$= \left(\boldsymbol{W}_{\mathrm{ZSL}} + \Delta\boldsymbol{W}_{\mathrm{ICL}}\right)\boldsymbol{q}. \tag{9}$$

Here, $\boldsymbol{W}_{\mathrm{ZSL}} = \boldsymbol{W}_V\boldsymbol{X}\left(\boldsymbol{W}_K\boldsymbol{X}\right)^T$ is the zero-shot activation from the static parameters of the model, in which $\boldsymbol{X}$ denotes the input representations of query tokens before the current one, $\boldsymbol{q}$. $\boldsymbol{X}'$ denotes the input representations of the demonstration tokens.

In summary, under the relaxed normalization setting, a pretrained LM acts as a meta-optimizer. Through forward computation, the LM generates meta-gradients from the demonstration examples, which are then applied to the original parameters via attention, culminating in the formation of the ICL inference capability.

This explanation provides an intuitive understanding of how the LM uses in-context examples, but it also highlights why ICL performance can be unstable. Specifically, meta-gradients derived from limited in-context examples may not fully capture the task and may not scale appropriately with the original parameters.

For this reason, we propose Iterative Vectors to extract meta-gradients—specifically, the activations induced by in-context examples—from the language model's inference process to enhance its accuracy and robustness. This would also allow us to apply these meta-gradients directly in future inference tasks, eliminating the need to compute them afresh with ICL each time a query is evaluated. However, before proceeding, it is necessary to establish the notations employed to evaluate activation vectors.

## 3.2 ACTIVATION VECTOR EVALUATION

We adhere to standard few-shot benchmarking protocols (Vinyals et al., 2016; Finn et al., 2017; Snell et al., 2017) to define the activation vector evaluation setting. For a given split of an $n$-way $k$-shot classification task $\mathcal{T} = \{\mathcal{T}_{\mathrm{train}}, \mathcal{T}_{\mathrm{val}}, \mathcal{T}_{\mathrm{test}}\}$, which comprises textual query-answer pairs $(x, y)$, an ICL *episode* [1] is sampled as:

$$E = [(x_1, y_1), \ldots, (x_{n\times k}, y_{n\times k}), (x_q, y_q)]. \tag{10}$$

---

[1]The term is borrowed from meta-learning, considering the meta-gradients at play.

Here, $(x_q, y_q)$ represents the query and its label, preceded by the $n \times k$ support examples. To avoid the impact of unbalanced samples, we uniformly sample $k$ examples from each of the $n$ classes and shuffle them to mitigate any bias arising from sample permutation. We maintain a record of the labels for each example, which can be accessed using $\text{Class}(x_i) \in \{1, 2, \ldots, n, q\}$.

The episode must first be converted into a pure text sequence before the language model $\text{LM}(\cdot)$ can process it. This conversion is handled by a *verbalizer*, which uses a predefined prompt template to instantiate the samples. The template contains two key components: the *input-output separator* that links a question with its answer, and the *example separator* that joins the given support set. To preserve the simplicity of the template, we have chosen to use one newline ($\backslash \text{n}$) for the input-output separator and three newlines for the example separator, as adopted in Min et al. (2022a).

When the language model $\text{LM}(\cdot)$ is provided with an episode $E$, it performs autoregressive inference on each of the tokens within the verbalized episode. The *clean* prediction of the language model is derived by applying the softmax function to the logits on the potential labels produced by the model, as expressed in the following equation:

$$\hat{y}_{\text{clean}} = \text{LM}(E). \tag{11}$$

In contrast, an *edited* run involves the use of an activation vector editor $f_{\text{edit}}$. The specific method of editing varies based on the chosen approach, and we express the general form as follows:

$$\hat{y}_{\text{edit}} = \text{LM}(E; f_{\text{edit}}(\mathbb{V}, \mathbb{P})), \tag{12}$$

which depends on the set of vectors $\mathbb{V}$ extracted by an *activation vector extractor*, $f_{\text{ext}}$, with hyperparameters $\mathbb{P}$:

$$\mathbb{V} = f_{\text{ext}}(\mathcal{T}_{\text{train}}; \mathbb{P}). \tag{13}$$

The extractor retrieves its target vectors $\mathbb{V}$ from $\mathcal{T}_{\text{train}}$ and identifies the optimal hyperparameters $\mathbb{P}^*$ from $\mathcal{T}_{\text{val}}$ by maximizing the metric $\text{M}$:

$$\mathbb{P}^* = \arg \max_{\mathbb{P}} \text{M}_{E \sim \mathcal{T}_{\text{val}}} (\hat{y}_{\text{edit}}, y_E) \tag{14}$$

$$\mathbb{V}^* = f_{\text{ext}}(\mathcal{T}_{\text{train}}; \mathbb{P}^*). \tag{15}$$

For single-token classification tasks, macro-F1, micro-F1, and weighted-F1 scores can serve as the metrics. The vectors $\mathbb{V}^*$ and the optimal hyperparameters $\mathbb{P}^*$ are then applied to the test set $\mathcal{T}_{\text{test}}$ to evaluate the final results $\text{M}_{E \sim \mathcal{T}_{\text{test}}} (\hat{y}_{\text{edit}}, y_q)$.

## 3.3 ITERATIVE VECTORS

We have demonstrated that attention layers significantly influence ICL, with demonstrations acting as meta-gradients to help the model adapt to the task during inference. We first specify the extractor, $f_{\text{ext}}$, for IV.

To extract the gradients, we construct two verbalized versions of a given $n$-way $k$-shot episode $E$. The first version, $E = [(x_1, y_1), \ldots, (x_{n \times k}, y_{n \times k}), (x_q, y_q)]$, is the standard shuffled verbalization, which serves as the complete episode. The second version, $E^0 = [(x_q, y_q)]$, is stripped of all demonstrations, resulting in a zero-shot query that provides no information about the task.

Input-output separators are responsible for generating the label words, which gather information and contribute to forming the final prediction (Wang et al., 2023), making the meta-gradients associated with them particularly important. Given their significance, during inference on the two versions, the extractor collects activations, $\text{Act}_l(x_i)$, for the input-output separator of the $i$-th example in the complete episode $E$, as well as $\text{Act}_l^0(x_q)$ of the query in the zero-shot query $E^0$, across each attention layer $l$ of the LM.

Subsequently, we subtract the zero-shot activations from the complete activations. Since there are no input-output separators for demonstrations in the zero-shot sequence, all activations from the complete episode use the activations on the input-output separator of the query as the subtrahend:

$$\Delta \text{Act}_l(x_i) = \text{Act}_l(x_i) - \text{Act}_l^0(x_q) \tag{16}$$

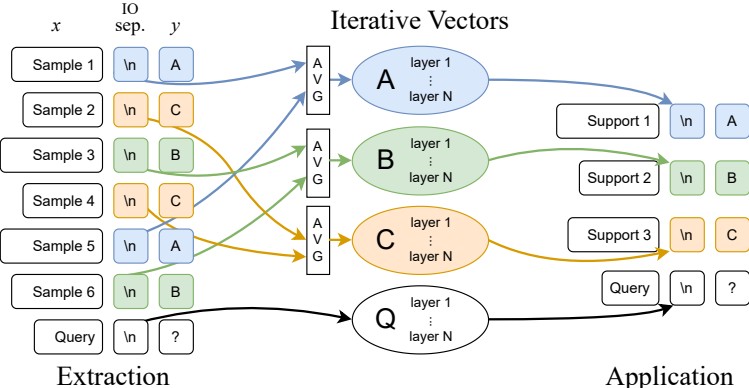

Figure 2: Illustration of the extraction and application of Iterative Vectors. For clarity, the subtraction and iterative updates have been omitted.

When $k > 1$, we average the activations for each class, resulting in $n$ vectors for each class, plus a vector for the final query:

$$\boldsymbol{v}_l^j = \frac{1}{|\mathbb{C}_j|} \sum_{i \in \mathbb{C}_j} \Delta \operatorname{Act}_l(x_i), \tag{17}$$

$$\boldsymbol{v}_l^q = \Delta \operatorname{Act}_l(x_q) = \operatorname{Act}_l(x_q) - \operatorname{Act}_l^0(x_q), \tag{18}$$

where $\mathbb{C}_j = \{i \mid \operatorname{Class}(x_i) = j\}$. This process yields the meta-gradients for a single episode

$$\mathbb{V}_l^E = \{\boldsymbol{v}_l^1, \boldsymbol{v}_l^2, \dots, \boldsymbol{v}_l^n, \boldsymbol{v}_l^q\}. \tag{19}$$

By averaging over the training set, a preliminary version of activation vectors can be obtained, as illustrated in Figure 2.

$$\mathbb{V}_l = \frac{1}{|\mathcal{T}|} \sum_{E \sim \mathcal{T}} \mathbb{V}_l^E \tag{20}$$

$$f'_{\text{ext}}(\mathcal{T}; \mathbb{P}) = \{\mathbb{V}_l; l \in \text{LM}\} \tag{21}$$

Next, to better utilize the forward pass computation, we propose to apply the vectors during the extraction phase, thus introducing the concept of *Iterative* Vectors. Specifically, we implement a batch-like update strategy to emulate standard batched gradient updates, a method commonly adopted to mitigate the instability associated with single-step gradients. After every $b$ episodes out of a total of $t$ extraction episodes, the IVs extracted are averaged and used as activation vectors to edit subsequent extractions dynamically.

$$\mathbb{V}^1 \leftarrow f'_{\text{ext}}(\mathcal{B}_1; \mathbb{P}), \qquad \mathbb{V}^{i+1} \xleftarrow[\text{while extracting}]{\text{edit with } \mathbb{V}^i} f'_{\text{ext}}(\mathcal{B}_{i+1}; \mathbb{P}) \tag{22}$$

$$f_{\text{ext}}(\mathcal{T}_{\text{train}}; \mathbb{P}) = \frac{1}{n} \sum_{i=1}^{n} \mathbb{V}^i \tag{23}$$

where $\mathcal{B}_i \sim \mathcal{T}_{\text{train}}$ represent the batches with size $|\mathcal{B}_i| = b$, and $n = t/b$ denotes the number of batched updates executed.

This process brings us to the definition of the editor, $f_{\text{edit}}$. For the $l$-th attention layer $\operatorname{Attn}_l(\cdot)$, we have the corresponding extracted IVs, $\mathbb{V}_l$. During inference, the editing is performed on each of the input-output separators with the IVs from their corresponding classes:

$$\operatorname{EditAttn}_l(x_i) = \operatorname{Attn}_l(x_i) + \alpha \times \boldsymbol{v}_l^{\operatorname{Class}(x_i)}. \tag{24}$$

Here, two additional hyperparameters are introduced: the extraction strength $\alpha_1$ and the inference strength $\alpha_2$, adopted during the iterative extraction and evaluation phases, respectively. In summary, the hyperparameters for IVs are $\mathbb{P} = \{k, b, \alpha_1, \alpha_2\}$.

Please refer to Appendix A for the pseudocode of our method, which provides a more detailed perspective on the methodology. Additionally, more information on hyperparameters can be found in Appendix F.

| Model | Method | abort. | agnews | athei. | clima. | emoti. | femin. | hate | hilla. | irony | offen. | senti. | sst5 | trec | **Avg**. |
|---|---|---|---|---|---|---|---|---|---|---|---|---|---|---|---|
| gpt-j-6b | Clean | 32.96 | 53.53 | 25.38 | **27.11** | 24.07 | 31.80 | 49.38 | 35.74 | **55.93** | 51.98 | 36.94 | 29.33 | 64.57 | 39.90 |
| | FV | **37.29** | 51.53 | **32.86** | 21.19 | 17.78 | 37.87 | 38.84 | 30.96 | 55.09 | 51.16 | **41.81** | 31.91 | 67.02 | 39.64 |
| | TV | 29.83 | **60.89** | 20.50 | 24.62 | 25.49 | 31.72 | **49.74** | 33.75 | 48.32 | 51.61 | 38.82 | 32.94 | 63.72 | 39.38 |
| | IV (Ours) | 36.06 | 56.13 | 32.05 | 19.23 | **32.70** | **38.20** | 47.30 | **40.68** | 54.65 | 46.32 | 33.17 | **39.07** | **67.32** | **41.76** |
| llama-2-7b | Clean | 27.52 | 61.94 | 22.13 | 28.60 | 54.45 | 29.27 | 53.27 | 29.42 | 58.65 | 51.86 | 38.96 | 28.93 | 74.93 | 43.07 |
| | FV | 25.11 | 67.56 | 14.58 | 23.70 | 58.66 | **31.01** | 52.57 | 32.26 | **60.44** | 54.89 | **42.40** | **30.89** | 71.29 | 43.49 |
| | TV | 27.91 | **72.11** | 21.75 | 31.98 | **59.37** | 29.56 | 50.08 | 29.54 | 50.21 | 52.00 | 41.64 | 29.94 | 74.77 | 43.91 |
| | IV (Ours) | **30.33** | 69.64 | **28.38** | **35.67** | 56.75 | 30.35 | **55.97** | 42.83 | 52.69 | 59.38 | 33.82 | 30.55 | **79.29** | **46.59** |
| llama-3.1-8b | Clean | 29.71 | 79.47 | 13.50 | 19.62 | 69.01 | 34.40 | 53.45 | 40.36 | 52.44 | 56.46 | 38.96 | 36.64 | 74.25 | 46.02 |
| | FV | 29.21 | 83.84 | 15.27 | 18.87 | 68.94 | 34.65 | 55.34 | 34.13 | **55.34** | **56.77** | **47.73** | 36.81 | 72.51 | 46.88 |
| | TV | **30.14** | 80.06 | 13.95 | 15.20 | 68.87 | 28.66 | 53.45 | **43.27** | 52.04 | 56.47 | 39.38 | 36.62 | 74.53 | 45.59 |
| | IV (Ours) | 29.81 | **87.13** | **23.49** | **23.01** | **69.73** | **36.84** | **58.82** | 40.34 | 50.21 | 55.29 | 42.45 | **41.50** | **75.63** | **48.79** |
| llama-2-13b | Clean | 34.96 | 76.23 | 27.11 | 20.96 | 61.89 | 37.13 | 53.83 | 45.53 | 45.53 | 55.17 | **60.34** | 38.77 | 38.66 | 76.01 | 48.20 |
| | FV | **36.55** | 77.37 | 27.25 | 19.71 | 66.73 | 43.35 | 50.57 | **51.16** | 51.26 | 58.94 | **46.15** | 42.72 | 72.57 | 49.56 |
| | TV | 34.71 | 76.28 | 27.24 | 30.88 | 63.27 | 31.87 | 52.63 | 45.03 | 54.98 | 60.14 | 37.82 | 37.98 | 77.05 | 48.45 |
| | IV (Ours) | 35.32 | **79.07** | **27.32** | **38.19** | **67.40** | **46.20** | **57.18** | 50.13 | **66.76** | 59.09 | 35.88 | **44.14** | **80.93** | **52.89** |

Table 1: Main experiment results with macro-F1 as the metric. "Clean" denotes a standard one-shot ICL result. The models are GPT-J-6B (Wang & Komatsuzaki, 2021), Llama 2 (Touvron et al., 2023) and Llama 3.1 (Dubey et al., 2024).

## 4 EXPERIMENTS

We apply our IVs to four popular models across 13 tasks. The results are presented in Table 1. Details of all the datasets used in this paper can be found in Appendix B, while additional results with the other two metrics are provided in Appendix C.

To provide additional proof of concept and comparative analysis, we include two recent activation vector proposals: Function Vectors (Todd et al., 2023) and Task Vectors (Hendel et al., 2023). Although these methods were not originally designed to operate under the ICL evaluation setting, we adapted them to utilize the training set by averaging the activations. We search over their respective hyperparameters as well as the extraction shot $k$ to ensure a fair comparison. Please refer to Appendix D for an overview of their designs.

During testing, the model cannot ascertain the true class distribution of the test set due to the few-shot setting, which is often imbalanced. Therefore, we adhere to one-shot during the main experiment, which supplies the model with minimal yet sufficient information through a set of uniformly distributed demonstration examples. A discussion on zero-shot sequences can be found in Appendix E.

We evaluate over 200 episodes for both extraction ($\mathcal{T}_{\text{train}}$) and hyperparameter search ($\mathcal{T}_{\text{val}}$). For the hyperparameters of IVs, we use a fixed iterative batch size of $b = 10$ and explore the extraction strength and inference strength $\alpha_1, \alpha_2 \in \{0.1, 0.3, 0.5, 0.7, 0.9\}$ for all tasks. Regarding the extraction shot $k$, we test $k \in \{1, 2, 3, 4\}$ for both TVs and IVs. However, due to their design (see Appendix D), FVs are excessively slow to extract, making it unfeasible to incorporate additional examples. Even when limited to $k = 1$, extracting FVs still takes about 20 times longer than extracting IVs. We present an example of the extraction time required in Table 2.

All experiments were conducted using a predetermined random seed (42) to mitigate selection bias. To ensure a robust representation of result distributions, the tests are averaged over a substantial number of episodes, namely 10,000. All experiments can be performed on a single Nvidia RTX A6000 GPU unless stated otherwise.

The results indicate that Iterative Vectors successfully achieve the goal, surpassing the baselines in most tasks as well as in the overall average. Task Vectors have demonstrated acceptable performance and can serve as a simple baseline for future research. Although Function Vectors achieve relatively better results than Task Vectors, their high search time presents significant challenges for optimization in practical ICL applications.

### 4.1 IVS SAVE INFERENCE TIME

All the aforementioned experiments require only a single demonstration during application, demonstrating that activation vectors can significantly reduce inference time. To highlight this point, we turn to the *emoji* dataset, a 20-class classification task (see Appendix B). Evaluating this dataset with multi-shot demonstrations would be exceedingly time-consuming due to the rapid increase in the length of the demonstration sequence.

| Setting | 1-shot | 2-shot | 3-shot | 4-shot | 1-shot + FV | 1-shot + TV | 1-shot + IV (ours) |
|---|---|---|---|---|---|---|---|
| **Macro-F1** | 9.13 | 12.90 | 12.64 | 13.11 | 10.77 | 10.30 | 12.90 |
| **Inference Time** (s) | 1374 | 2434 | 3426 | 4506 | 1389 | 1384 | 1452 |
| **Extraction Time** (min) | - | - | - | - | 438.3 | 14.58 | 23.75 |

Table 2: Clean and activation vector results on the *emoji* dataset with model Llama-2-7b. Inference time measurements are based on 10,000 episodes, while extraction is based on 200 episodes.

| Dataset | 2-shot | | | 3-shot | | | 4-shot | | | 5-shot | | |
|---|---|---|---|---|---|---|---|---|---|---|---|---|
| | Clean | +IV | Diff | Clean | +IV | Diff | Clean | +IV | Diff | Clean | +IV | Diff |
| AG News | 76.86 | 79.94 | +3.08 | 80.55 | 82.49 | +1.94 | 82.12 | 84.82 | +2.70 | 82.47 | 85.84 | +3.37 |
| Rotten Tomatoes | 70.28 | 87.50 | +17.22 | 78.97 | 90.57 | +11.60 | 83.74 | 90.74 | +7.00 | 87.80 | 91.48 | +3.68 |

Table 3: Multi-shot clean and IV results using the Llama-2-7b model. The displayed metric is macro-F1.

We apply IV on this dataset and further fix the extraction shot at $k = 1$ rather than exploring the range $k = \{1, 2, 3, 4\}$ to further minimize the time required for hyperparameter search. The results, presented in Table 2, clearly show that IVs substantially enhance performance with minimal time expenditure, in stark contrast to higher-shot ICL cases, which required significantly more time.

## 4.2   IVS SCALE WITH IN-CONTEXT DEMONSTRATIONS

One might wonder why activation vectors are not applied to higher-shot settings. The primary reason is that a key objective of using activation vectors is to reduce the inference time associated with higher-shot scenarios. Nonetheless, we conducted experiments to evaluate their performance with longer demonstrations.

For this study, we have chosen the *AG News* and *Rotten Tomatoes* datasets. This selection is based on the observation that the language model under evaluation demonstrates progressively improved performance as the number of examples increases, as illustrated in Table 3. Consequently, this poses a more substantial challenge for the IVs to improve upon. However, the results demonstrate that IVs scale effectively with the number of demonstration shots preceding the query. This suggests that IVs can offer advantages even when initial performance levels are already high, and they integrate seamlessly with the ICL framework.

In addition, one could contemplate a similar challenge using larger models. The results are comparable; please refer to Table 8, where the improvement of IVs is once again evident with Llama-2-70b.

## 4.3   IVS IMPROVE WITH INCREASED EXTRACTION EPISODES

An important aspect to consider is the number of examples required for IVs to function effectively. We conduct an experiment to test various numbers of extraction episodes, which in turn controls the number of examples used to extract the IVs.

Another critical aspect is the stability of IVs when extracted from different numbers of episodes. To evaluate this, we utilized hyperparameters obtained from prior searches in the main experiment ($k = 4$, fixed $b = 10$, $\alpha_1 = 0.3$, $\alpha_2 = 0.5$), rather than optimizing hyperparameters for each different episode count. The results are presented in Table 4.

The data shows that, although there are some fluctuations when the episode number is small, IVs extracted from more than two episodes consistently enhance performance (higher than the clean performance 62.15), even with fixed, potentially suboptimal hyperparameters. Overall, performance improves as the number of examples increases, demonstrating IVs' ability to extract and utilize a greater number of examples, thereby exceeding the conventional limits of ICL.

| Episodes | 1 | 2 | 3 | 5 | 10 | 20 | 30 | 50 | 100 | 150 | 200 |
|---|---|---|---|---|---|---|---|---|---|---|---|
| Macro-F1 | 40.64 | 54.44 | 62.72 | 66.17 | 64.27 | 63.01 | 65.05 | 66.77 | 68.14 | 69.71 | 69.62 |

Table 4: IV results with different number of extraction episodes, using a fixed set of hyperparameters. The model utilized is Llama-2-7b, and the dataset is AG News.

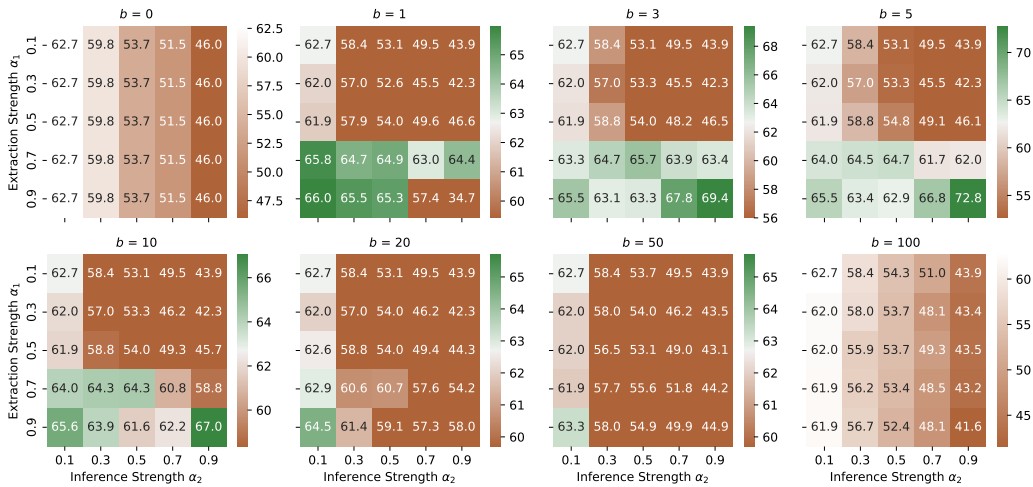

Figure 3: Ablation study on the hyperparameters. The model utilized is Llama-2-7b, and the dataset evaluated is the validation split of AG News, with macro-F1 serving as the metric. Note that $b = 0$ indicates no iterative refining and batching.

## 4.4 ABLATION STUDY

We present an ablation study on the hyperparameters of our method. In all previous experiments, the extraction batch size is fixed at $b = 10$. In this study, we vary this parameter to observe its impact on other hyperparameters. The results are presented in Figure 3.

To examine the hyperparameter search process, we focus on the validation phase, during which the optimal hyperparameters are determined. When $b = 0$, the extracted vectors are not reintroduced into the model, resulting in poor performance compared to other cases. Without editing during extraction, the extraction strength $\alpha_1$ also becomes non-reactive. When $b = 1$, even though effective batching is not present, reintroducing the extracted vectors into the model for refinement results in a significant performance boost. This underscores the importance of *Iterative* Vectors.

As the batch size increases, the optimal hyperparameter pairs initially emerge in the bottom left corner, characterized by a high extraction strength $\alpha_1$ and a low inference strength $\alpha_2$. This suggests that with a small batch size, the extracted vectors lack stability, making them unsuitable for inference. As the batch size continues to grow, the optimal inference strength $\alpha_2$ also increases, reaching an effective combination. However, once the batch size becomes excessively large, it adversely affects the hyperparameters.

These interactions underscore the importance and contribution of each hyperparameter to the overall methodology. For a more comprehensive discussion, including guidance on tuning them, please refer to Appendix F.

## 5 CONCLUSION

In our study, we have derived the Iterative Vectors (IVs) from an intuitive theoretical framework, defined the evaluation protocols and subsequently conducted a series of experiments. Despite IVs' simplicity, the results obtained are highly encouraging, indicating that activation vectors show significant potential for further exploration.

## LIMITATIONS

This study examines the application of Iterative Vectors in the context of one-shot examples as a compromise between inference time and in-context information. Although applying IVs to zero-shot inference would be more efficient, a computational sequence of insufficient length might hinder the model's ability to effectively solve the given task. For additional discussion, please refer to Appendix E.

We have opted for classification tasks wherein a single output token is sufficient to distinguish between the classes. The development and application of activation vectors in more complex tasks, as well as in generative tasks, represent areas for future investigation. Nevertheless, it is worth noting that the concept of IVs and the associated evaluation protocol can potentially be expanded to encompass these more advanced applications.

## REPRODUCIBILITY STATEMENT

We have provided a comprehensive set of pseudocode in Appendix A, which is crucial for implementing our method. The datasets used are detailed in Appendix B.

Furthermore, we plan to release the complete code repository necessary for reproducing all of our experiments to promote transparency and facilitate future research endeavors.

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

## A PSEUDOCODE

We first define three utility functions used for the extraction and application of IVs, as indicated in Algorithm 1. Subsequently, we outline the procedures for IV extraction in Algorithm 2 and evaluation in Algorithm 3.

Regarding our hyperparameters, please refer to the extraction shot $k$, batch size $b$, and strength $\alpha_1$, as specified in Algorithm 2. Additionally, consult the inference strength, denoted as $\alpha_2$, in Algorithm 3.

## B DATASETS

A full list of all datasets utilized in this research, along with their corresponding access labels, is detailed in Table 5. The datasets are obtained from HuggingFace (Lhoest et al., 2021).

AG News (Zhang et al., 2015) is a subdataset of AG's corpus of news articles constructed by assembling titles and description fields of articles from the 4 largest classes ("World", "Sports", "Business", "Sci/Tech") of AG's Corpus.

TweetEval (Barbieri et al., 2020) introduces an evaluation framework consisting of a series of Twitter-specific classification tasks. We selected all single-token classification tasks from the dataset.

---

**Algorithm 1** Episodic Functions

---

1: **function** EXTRACT(sequence)                                   ▷ Extracts activations from the LM
2:     $v \leftarrow \varnothing$
3:     **run** LM(sequence) **with**                      ▷ Hook into the LM with the following operations
4:         **for** each layer in LM **do**                                                              ▷
5:             $p \leftarrow$ the position of the input-output separator after the query
6:             $v \leftarrow v \cup \{\text{Attn}[p]\}$                      ▷ Store the activation of each attention layer
7:         **end for**
8:     **end run**
9:     **return** $v$
10: **end function**
11: **function** APPLY(sequence, $\mathbb{V}$, $\alpha$)                       ▷ Apply IV to LM inference process
12:     **run** logits $\leftarrow$ LM(sequence) **with**
13:         **for** each layer in LM **do**
14:             **for** each support sample in sequence **do**
15:                 $p \leftarrow$ the position of the input-output separator after the sample
16:                 $c \leftarrow$ the class of the sample
17:                 $\text{Attn}[p] \leftarrow \text{Attn}[p] + \alpha \times \mathbb{V}[c]$     ▷ Edit the separators in the support sequence...
18:             **end for**
19:             $p \leftarrow$ the position of the input-output separator after the query
20:             $\text{Attn}[p] \leftarrow \text{Attn}[p] + \alpha \times \mathbb{V}[\text{QUERY}]$                      ▷ ...as well as the query
21:         **end for**
22:     **end run**
23:     **return** logits
24: **end function**
25: **function** APPLYANDEXTRACT(sequence, $\mathbb{V}$, $\alpha$)                     ▷ Apply the IV during extraction
26:     $v \leftarrow \varnothing$
27:     **run** LM(sequence) **with**
28:         **for** each layer in LM **do**
29:             **if** $\mathbb{V} \neq \varnothing$ **then**                       ▷ The first batch does not have $\mathbb{V}$ for editing
30:                 **for** each support sample in sequence **do**
31:                     $p \leftarrow$ the position of the input-output separator after the sample
32:                     $c \leftarrow$ the class of the sample
33:                     $\text{Attn}[p] \leftarrow \text{Attn}[p] + \alpha \times \mathbb{V}[c]$                              ▷ Edit (support)
34:                 **end for**
35:                 $p \leftarrow$ the position of the input-output separator after the query
36:                 $\text{Attn}[p] \leftarrow \text{Attn}[p] + \alpha \times \mathbb{V}[\text{QUERY}]$                              ▷ Edit (query)
37:             **end if**
38:             $p \leftarrow$ the position of the input-output separator after the query
39:             $v \leftarrow v \cup \{\text{Attn}[p]\}$                              ▷ Extract and append to list
40:         **end for**
41:     **end run**
42:     **return** $v$
43: **end function**

---

The Rotten Tomatoes dataset (Pang & Lee, 2005) is a collection of movie reviews and ratings from the Rotten Tomatoes website, often used for sentiment analysis and natural language processing tasks.

The SST5 dataset, derived from the Stanford Sentiment Treebank (Socher et al., 2013), is a collection of movie reviews annotated with fine-grained sentiment labels, offering a five-class sentiment classification task ranging from very negative to very positive.

Text Retrieval Conference Question Answering (TrecQA) (Wang et al., 2007) is a dataset created from the TREC-8 (1999) to TREC-13 (2004) Question Answering tracks.

Our few-shot evaluation methodology employs episodic sampling to regulate the duration of both extraction and inference processes, rather than relying solely on the absolute number of samples.

---

**Algorithm 2** Extraction

---

**Require:** extraction shot: $k$, extraction batch size: $b$, extraction strength: $\alpha_1$
**Ensure:** extracted Iterative Vector: $\mathbb{V}$

1: $\mathbb{V} \leftarrow \varnothing$                 ▷ Initialize the variable to store the IV
2: ivs $\leftarrow \varnothing$              ▷ An empty list to store IV for each episode
3: **for** the $i$-th episode **do**
4:      support, query $\leftarrow$ RANDOMEPISODE($k$)       ▷ Sample a $k$-shot episode
5:      order, support $\leftarrow$ SHUFFLE(support)      ▷ Shuffle and remember the classes
6:      sq_seq $\leftarrow$ VERBALIZE(support $\oplus$ query)      ▷ Convert to few-shot prompt
7:      q_seq $\leftarrow$ VERBALIZE(query)      ▷ Convert to zero-shot prompt
8:      sq_vec $\leftarrow$ APPLYANDEXTRACT(sq_seq, $\mathbb{V}$, $\alpha_1$)
9:      q_vec $\leftarrow$ EXTRACT(q_seq)
10:     **for** each class of the task **do**
11:        $p \leftarrow$ the position(s) where order is equal to class      ▷ Collect by each class
12:        $v[\text{class}] \leftarrow$ MEAN(sq_vec[$p$] $-$ q_vec)      ▷ Average over shots
13:     **end for**
14:     $v[\text{QUERY}] \leftarrow$ sq_vec[QUERY] $-$ q_vec      ▷ Collect the query as well
15:     ivs $\leftarrow$ ivs $\cup \{v\}$      ▷ Append the current episode's IV to the list
16:     **if** $i \bmod b = 0$ **then**      ▷ Check if the current episode is a multiple of batch size
17:        $\mathbb{V} \leftarrow$ MEAN(ivs)      ▷ Update the IV to apply as the average over episodes
18:     **end if**
19: **end for**

---

**Algorithm 3** Evaluation

---

**Require:** evaluation shot $k'$, extracted Iterative Vector: $\mathbb{V}$, inference strength: $\alpha_2$
**Ensure:** classification labels: results

1: results $\leftarrow \varnothing$          ▷ An empty list to store results for each episode
2: **for** the $i$-th episode **do**
3:      support, query $\leftarrow$ RANDOMEPISODE($k'$)      ▷ Sample an episode, typically with $k' = 1$
4:      support $\leftarrow$ SHUFFLE(support)      ▷ Shuffle to avoid patterned few-shot sequence
5:      sq_seq $\leftarrow$ VERBALIZE(support $\oplus$ query)      ▷ Convert to prompt
6:      logits $\leftarrow$ APPLY(sq_seq, $\mathbb{V}$, $\alpha_2$)      ▷ Run the LM with editing
7:      results $\leftarrow$ results $\cup \{$ARGMAX(logits[labels])$\}$      ▷ Calculate the classification result
8: **end for**

---

Consequently, not all available samples are utilized during the experimental procedures. This aspect underscores an additional dimension of efficiency inherent in activation vectors.

## C ADDITIONAL RESULTS

We present the results of our main experiment on the other two metrics, namely micro-F1 and weighted-F1, derived from our main experiment, in Table 6 and Table 7, respectively.

According to these evaluation criteria, IV outperforms both FV and TV in the majority of tasks, consistently achieving a higher average score. The only exception occurs in the GPT-J-6B and micro-F1 setting (Table 6), where FV demonstrates superior performance. We hypothesize that this result indicates a bias of FV towards the majority classes in this specific model. This bias leads to an increased micro-F1 score; however, it causes the macro-F1 score to drop below the clean baseline.

An additional experiment was conducted utilizing the Llama-2-70b model. Due to our computational budget constraints, it was not feasible to complete all tasks with a model of this scale. Therefore, we opted to conduct a multi-shot experiment, as described in Section 4.2 (Table 3), to more effectively showcase the efficacy of IV. The results are presented in Table 8.

| Name | Abbr. Used | Huggingface Label |
|------|-----------|-------------------|
| Abortion | abor. | tweet_eval/stance_abortion |
| AG News | agnews | ag_news |
| Atheism | athe. | tweet_eval/stance_atheism |
| Climate | clim. | tweet_eval/stance_climate |
| Emoji | - | tweet_eval/emoji |
| Emotion | emot. | tweet_eval/emotion |
| Feminist | femi. | tweet_eval/stance_feminist |
| Hate | hate | tweet_eval/hate |
| Hillary | hill. | tweet_eval/stance_hillary |
| Irony | irony | tweet_eval/irony |
| Offensive | offe. | tweet_eval/offensive |
| Rotten Tomatoes | - | rotten_tomatoes |
| Sentiment | sent. | tweet_eval/sentiment |
| SST 5 | sst5 | SetFit/sst5 |
| TREC | trec | trec |

Table 5: The datasets and tasks employed, along with their corresponding abbreviations used in the result tables, and their respective labels as hosted on Hugging Face.

| Model | Task | abort. | agnews | athei. | clima. | emoti. | femin. | hate | hilla. | irony | offen. | senti. | sst5 | trec | Avg. |
|-------|------|--------|--------|--------|--------|--------|--------|------|--------|-------|--------|--------|------|------|------|
| gpt-j-6b | Clean | 39.17 | 57.97 | 30.49 | 30.92 | 31.91 | 37.70 | 49.39 | 40.33 | **59.86** | **63.22** | 38.73 | 32.62 | 68.23 | 44.66 |
| | FV | 51.93 | 55.39 | **45.81** | 24.89 | 29.62 | **54.20** | 45.48 | **58.97** | 57.30 | 58.25 | **41.77** | 37.37 | **69.70** | **48.51** |
| | TV | 51.52 | **65.86** | 23.72 | **32.84** | 32.85 | 37.64 | **49.74** | 37.89 | 48.32 | 60.05 | 40.23 | 35.60 | 64.75 | 44.69 |
| | IV (Ours) | **60.02** | 61.30 | 44.59 | 20.49 | **37.36** | 49.05 | 48.32 | 55.29 | 56.30 | 46.94 | 34.48 | **40.08** | 67.32 | 47.81 |
| llama-2-7b | Clean | 28.69 | 63.40 | 24.90 | 34.88 | 57.31 | 30.25 | 53.64 | 30.05 | 62.22 | 53.67 | 40.02 | 43.08 | **77.33** | 46.11 |
| | FV | 30.25 | 69.56 | 18.50 | 25.49 | **62.91** | 36.07 | 57.16 | 35.29 | **63.83** | **63.95** | **46.44** | 45.22 | 75.54 | 48.48 |
| | TV | 29.31 | **72.97** | 24.50 | 62.14 | 62.52 | 30.47 | 50.09 | 30.14 | 52.86 | 53.53 | 41.07 | 43.28 | 77.10 | 48.46 |
| | IV (Ours) | **35.88** | 72.45 | **39.17** | 58.46 | 58.96 | **40.03** | 58.46 | 48.83 | 53.01 | 63.59 | 36.25 | **46.67** | 76.83 | **52.97** |
| llama-3.1-8b | Clean | 39.18 | 80.64 | 18.14 | 21.26 | 74.06 | 47.17 | 53.66 | 48.14 | 53.96 | 60.12 | 39.01 | 45.25 | 69.69 | 50.02 |
| | FV | 41.93 | 84.31 | 21.15 | 20.47 | 74.35 | 51.76 | 55.45 | 44.08 | **56.06** | **69.89** | **48.32** | 42.43 | 68.20 | 52.18 |
| | TV | 39.07 | 81.12 | 18.55 | 20.21 | 74.47 | 40.21 | 53.47 | 50.33 | 53.67 | 60.35 | 39.13 | 43.04 | 69.62 | 49.48 |
| | IV (Ours) | **44.25** | **87.30** | **36.33** | **22.33** | **77.70** | **56.57** | **58.84** | **56.07** | 52.23 | 69.20 | 42.83 | **48.85** | **70.24** | **55.60** |
| llama-2-13b | Clean | 52.57 | 77.96 | 42.78 | 20.36 | 65.42 | 55.94 | 54.00 | 56.83 | 55.19 | 63.56 | 41.41 | 44.44 | **78.56** | 54.54 |
| | FV | 53.16 | 78.81 | **48.92** | 19.57 | 69.99 | **64.96** | **58.94** | 62.25 | 52.32 | 70.70 | **47.87** | **49.19** | 76.58 | 57.94 |
| | TV | 51.34 | 78.07 | 43.22 | 49.38 | 67.27 | 47.60 | 53.22 | 56.05 | 55.05 | 62.82 | 39.70 | 43.86 | 76.16 | 55.67 |
| | IV (Ours) | **55.67** | **80.33** | 46.74 | **65.56** | **71.03** | 58.84 | 58.67 | **63.13** | **66.96** | **73.80** | 36.74 | 47.90 | 77.47 | **61.76** |

Table 6: Main experiment results with micro-F1 as the metric. "Clean" denotes a standard one-shot ICL result.

## D  COMPARISON OF METHODOLOGIES

We will begin with an introduction to the motivation and functioning of FV and TV. Following this, we will offer comprehensive comparisons from various perspectives.

**Function Vectors.**   Function Vectors (Todd et al., 2023) are inspired by the observation that incorporating activations extracted from few-shot tasks on the last token at specific layers can prompt an LM to execute a task when applied to an unseen zero-shot prompt. To distill a more effective hidden-state representation, the researchers limit their investigation to attention heads. This decision is based on the heuristic that attention heads are the components used by transformers to transfer information across different token positions. The researchers aim to identify attention heads that have a causal influence on predicting the desired label for a given task. The method for calculating this causal effect is outlined as follows:

1. Compute the average activation $\bar{a}_{\ell j}^t$ of each attention head $j$ at layer $\ell$ over task $t$.

2. Feed the ICL prompt $\tilde{p}_i^t$ with shuffled labels into model $f$, and calculate the probability assigned to the target label $f(\tilde{p}_i^t)$.

3. Use one $\bar{a}_{\ell j}^t$ to replace the activation of its corresponding attention head, conducting a separate run for each instance. Subsequently, compute the edited probability for the target label again as $f(\tilde{p}_i^t | a_{\ell j} = \bar{a}_{\ell j}^t)$.

| Model | Task | abort. | agnews | athei. | clima. | emoti. | femin. | hate | hilla. | irony | offen. | senti. | sst5 | trec | **Avg.** |
|---|---|---|---|---|---|---|---|---|---|---|---|---|---|---|---|
| gpt-j-6b | Clean | 42.61 | 53.69 | 34.82 | **34.83** | 22.48 | 40.34 | 49.46 | 42.14 | **58.64** | 62.47 | 33.50 | 31.82 | 68.12 | 44.22 |
| | FV | 52.83 | 51.62 | **50.11** | 31.38 | 17.29 | **52.93** | 35.96 | 47.47 | 57.23 | 59.41 | **39.82** | 35.19 | **69.86** | 46.24 |
| | TV | 49.48 | **61.07** | 26.01 | 34.19 | 22.74 | 40.30 | **49.79** | 39.49 | 48.21 | 60.68 | 34.78 | 35.52 | 65.18 | 43.65 |
| | IV (Ours) | **56.37** | 56.16 | 48.98 | 15.48 | **33.59** | 50.39 | 46.26 | **52.34** | 56.49 | 48.88 | 32.98 | **40.08** | 68.38 | **46.64** |
| llama-2-7b | Clean | 30.58 | 62.03 | 27.50 | 38.72 | 57.45 | 31.75 | 53.83 | 27.79 | 61.15 | 56.07 | 35.33 | 34.46 | 77.58 | 45.71 |
| | FV | 31.40 | 67.69 | 16.00 | 25.62 | **62.86** | 38.41 | 54.68 | 33.09 | **62.93** | 63.85 | 35.83 | 36.79 | 77.29 | 46.65 |
| | TV | 31.43 | **72.23** | 27.39 | **60.09** | 62.70 | 32.06 | 50.00 | 27.66 | 52.57 | 55.85 | **39.36** | 35.39 | 77.27 | 48.00 |
| | IV (Ours) | **38.90** | 69.75 | **44.22** | 59.10 | 59.02 | **41.32** | **57.46** | 50.01 | 51.86 | **65.18** | 27.70 | **36.94** | **78.22** | **52.28** |
| llama-3.1-8b | Clean | 40.92 | 79.57 | 15.32 | **13.97** | 73.77 | 47.66 | 53.04 | 48.62 | 50.70 | 62.16 | 36.04 | 40.44 | 70.66 | 48.68 |
| | FV | 43.03 | 83.91 | 20.32 | 10.22 | 74.01 | 50.30 | 55.02 | 43.71 | **54.11** | 67.33 | **44.67** | 38.50 | 70.74 | 50.45 |
| | TV | 41.06 | 80.17 | 16.45 | 9.35 | 73.86 | 41.34 | 53.33 | 51.20 | 50.23 | 62.30 | 36.09 | 39.41 | 70.65 | 48.11 |
| | IV (Ours) | **44.98** | **87.18** | **39.73** | 11.41 | **76.67** | **53.66** | **58.70** | **54.28** | 48.05 | 66.34 | 38.88 | **44.27** | **72.86** | **53.62** |
| llama-2-13b | Clean | 51.80 | 76.36 | 45.57 | 19.77 | 65.73 | 53.00 | 53.46 | 55.25 | 54.99 | 65.44 | 33.47 | 41.63 | 79.10 | 53.51 |
| | FV | 52.92 | 77.47 | **49.87** | 22.99 | 70.76 | **60.23** | 53.47 | **60.28** | 49.71 | 68.68 | **41.76** | 46.51 | 78.98 | 56.43 |
| | TV | 51.32 | 76.43 | 45.95 | 51.92 | 67.44 | 46.91 | 51.91 | 54.67 | 54.63 | 64.78 | 32.12 | 41.10 | 77.07 | 55.10 |
| | IV (Ours) | **53.93** | **79.17** | 48.74 | **63.85** | **71.40** | 59.55 | **58.32** | 58.96 | **67.31** | **69.96** | 35.51 | **46.82** | **79.27** | **60.98** |

Table 7: Main experiment results with weighted-F1 as the metric. "Clean" denotes a standard one-shot ICL result.

| Dataset | 1-shot | | | 2-shot | | | 3-shot | | | 4-shot | | |
|---|---|---|---|---|---|---|---|---|---|---|---|---|
| | Clean | +IV | Diff | Clean | +IV | Diff | Clean | +IV | Diff | Clean | +IV | Diff |
| AG News | 86.96 | 88.17 | +1.21 | 87.99 | 89.04 | +1.05 | 87.87 | 88.84 | +0.97 | 89.01 | 89.32 | +0.31 |
| Rotten Tomatoes | 82.24 | 91.52 | +9.28 | 91.29 | 92.38 | +1.09 | 92.39 | 93.13 | +0.74 | 92.50 | 92.69 | +0.19 |

Table 8: Multi-shot clean and IV results using the Llama-2-70b model. The displayed metric is macro-F1. Conducted on 3 Nvidia RTX A6000 GPUs.

4. The *causal indirect effect* on task $t$ and the shuffled prompt $\tilde{p}_i^t$ is calculated as

$$\text{CIE}(a_{\ell j} \mid \tilde{p}_i^t) = f(\tilde{p}_i^t \mid a_{\ell j} := \bar{a}_{\ell j}^t) - f(\tilde{p}_i^t). \tag{25}$$

5. The *average indirect effect* is the average of the CIE across all tasks and prompts:

$$\text{AIE}(a_{\ell j}) = \frac{1}{|\mathcal{T}|} \sum_{t \in \mathcal{T}} \frac{1}{|\tilde{P}_t|} \sum_{\tilde{p}_i^t \in \tilde{P}_t} \text{CIE}(a_{\ell j} \mid \tilde{p}_i^t). \tag{26}$$

6. Gather the attention heads with highest AIE over all layers to serve as the activation source, forming set $\mathcal{A}$.

The researchers represent the contribution of $\mathcal{A}$ as a single vector by taking the sum of their average outputs, over a task, which is called a Function Vector for task $t$:

$$v_t = \sum_{a_{lj} \in \mathcal{A}} \bar{a}_{lj}^t. \tag{27}$$

To utilize FV, add it to the activation of the final token at a designated layer as the model processes a prompt.

One significant issue with FV is that it necessitates an extensive search through all attention heads of every layer, posing considerable scaling challenges as the model size grows. Theoretically, aside from the extraction time attributed to the extraction shot $k$, the extraction time of FV increases with an additional complexity of $O(E \times L \times H)$. Here, $E$ represents the number of extraction episodes, $L$ denotes the layer count of the LM, and $H$ is the number of attention heads in each layer. For example, GPT-J-6B has a total of 448 heads, while Llama-2-13B has 1600. This increase alone more than triples the time required to extract the FVs, not to mention the slower computation resulting from a longer prompt and a larger model size.

In contrast, Task Vector and our Iterative Vector do not encounter this issue and scale smoothly with larger models. During our experiments, we had to restrict the extraction shot $k$ for FV to maintain practical search times and ensure fairness across all evaluated methods, as mentioned in Section 4.

**Task Vectors.** Task Vectors (Hendel et al., 2023) offer a mechanistic perspective on ICL. This approach conceptualizes ICL as a two-step process: first, a parameter vector $\theta$ is computed from the training sample, which is subsequently used to apply the "rule" defined by the vector to the query $x$.

There are many possible realizations of the above framework. The researchers presume that a simple way for a transformer to achieve this is for the initial $L$ layers to compute $\theta$. The remaining layers would take $\theta$ and $x$ as inputs to generate an output.

This provides a straightforward method to extract the language model's knowledge of a task and subsequently apply it. The process involves performing a forward pass of the transformer and utilizing the previously extracted $\theta$ to patch the $L$-th layer of the final token.

However, the boundary that separates this artificially divided two-stage process in the LM remains unclear and needs to be selected through empirical searching.

**Comparison with Iterative Vectors.** The theoretical attributes of our methodology, in comparison to the baseline models, are as follows:

- FV and TV utilize their experiments to validate their respective hypotheses, rather than basing their methods on theoretical foundations.
- Consequently, their editing processes are heuristic and rely on intuition.
- Our proposed method is grounded in the meta-gradients derived from the demonstrations through the computation of the attention modules within the model.
- This approach not only identifies where to make edits (the attention layers) but also specifies how to perform the edits (by performing meta-gradient updates via adding to the activations).

The extraction and editing process differs considerably for each method, as illustrated below:

- FV examines all attention heads and aggregates the activations of the top-performing ones to obtain the vectors, which is highly time-consuming.
- TV simply identifies the optimal layer for the extraction and application of vectors.
- IV processes the activations from different classes separately, conducting aggregation and application based on this separation. We also propose iterative updates and batched extraction for meta-gradients, which have been proven to significantly enhance performance.

The hyperparameters specific to each method (instead of the evaluation framework) are as follows:

- FV: the count of top heads $|\mathcal{A}|$ and the layer to apply the vector.
- TV: the layer to apply the vector.
- IV: extraction strength $\alpha_1$, inference strength $\alpha_2$, and iterative batch size $b$.

Please refer to Appendix F for a more detailed discussion on the hyperparameters of IV.

As a side note, we can see from the comparisons above that there is considerable flexibility in the design of activation vectors. We hope that our efforts will serve as a catalyst for further exploration and advancement in this line of inquiry, ultimately unlocking the full potential of activation vectors.

## E    CONCERNING ZERO-SHOT SEQUENCES

In both the FV and TV papers (Todd et al., 2023; Hendel et al., 2023), the vectors are utilized on zero-shot sequences. This aims to demonstrate the effectiveness of activation vectors in guiding the model as expected. The results confirm this: zero-shot sequences with activation vectors differ significantly from clean zero-shot runs. However, there remains a noticeable gap between zero-shot applications and standard few-shot ICL performance, which appears difficult to bridge. For instance, in Figure 4 of the TV paper, all FV runs fall behind the few-shot runs across all models, despite the tasks being simple synthetic ones.

Previous research has suggested reasons that may account for this disparity. Feng et al. (2023) provide fundamental impossibility results, indicating that language models cannot solve increasingly complex tasks in a single generation step. If we view the demonstration sequence as an extension of the inference steps generated by the LM—since the model treats all previous tokens equally,

whether generated or provided—then without demonstrations, the LM's capabilities are significantly impaired. A zero-shot attempt might not provide adequate computation for the language model to effectively address a given task. Consequently, it might be overly optimistic to expect activation vectors to circumvent all necessary computations.

Furthermore, Min et al. (2022b) demonstrated the importance of informing the LM about the label space of the current task to enhance ICL performance. In a zero-shot scenario, the model might struggle to focus its classification ability on the desired label, instead distributing it across the entire vocabulary space, as noted by Holtzman et al. (2021). This adds an extra burden for the model to extract meta-gradients and adjust accordingly.

Our early experiments on real-world tasks also confirmed that activation vectors do not perform well in a zero-shot setting. While there are some improvements, they remain inferior compared to the results achieved with even a one-shot approach. For synthetic experiments, these results may be adequate; however, to make activation vectors effective for practical applications, we must achieve better outcomes.

Consequently, we have decided to focus on enhancing few-shot performance rather than zero-shot. Table 2 of the FV paper offers a compelling insight: FV is applied not only to zero-shot sequences but also to "uninformative" sequences, which are essentially few-shot sequences with shuffled labels. These shuffled sequences nearly double the performance compared to their zero-shot counterparts on synthetic tasks, prompting us to begin our investigation from this point. However, since using a shuffled sequence is not meaningful for our purposes, we employ a correct one-shot sequence instead. The advantages of this approach include a basic guarantee of performance, along with the presence of input-output separators in the support samples, which further facilitate the application of the vectors.

Nonetheless, we hope our research will enhance future studies on activation vectors, enabling them to more effectively address the zero-shot scenario. This would represent a significant, albeit challenging, advancement.

## F    HYPERPARAMETERS OF IV

In this paper, we introduce four hyperparameters: the extraction shot $k$, the extraction batch size $b$, the extraction strength $\alpha_1$, and the inference strength $\alpha_2$. These notations have been used consistently throughout the paper, including in formulas, pseudocode, and explanations. We now provide a detailed discussion of each hyperparameter and its function, followed by a guide on how to tune them effectively.

**The extraction shot $k$ controls the number of samples in a sequence during the extraction process.** This originates from the definition of an $n$-way $k$-shot episode (Eq. 10). During extraction, a longer support sequence may enhance the model's understanding of the task, thereby producing higher-quality meta-gradients. However, since adding more samples does not always improve performance, and a larger $k$ increases extraction time, we propose optimizing this hyperparameter through a search process.

**The extraction batch size $b$ serves to replicate a typical batch size used during standard training.** As implemented in Algorithm 2, the preliminary vectors extracted are averaged every $b$ episodes to form the Iterative Vectors, which are subsequently incorporated into the extraction process. Since we are extracting meta-gradients to be applied to the model's hidden states, we propose utilizing them during the extraction process rather than waiting for its completion. Iterative refinement enables each layer in the language model to be guided by meta-gradients, thereby influencing subsequent layers to generate enhanced representations. This process aids in contrasting the zero-shot sequence and provides improved meta-gradients.

In Section 4.4, we analyzed the impact of varying the parameter $b$ on performance, as well as its influence on other parameters. We found that an appropriate batch size can significantly enhance performance.

**The extraction strength $\alpha_1$ denotes the magnitude with which meta-gradients are applied during iterative extraction.** Similarly, the inference strength $\alpha_2$ represents the magnitude with which meta-gradients are applied during evaluation. These two parameters share the same notation because they fundamentally represent the same concept, albeit applied in different phases.

In the application of vectors, all methods evaluated in this paper utilize vector addition. However, the meta-gradients may not scale properly with the original parameters. Therefore, we propose scaling them before incorporating them into the hidden states, a consideration not derived from nor addressed in previous methods. During the iterative extraction phase, the scaling constant is $\alpha_1$, whereas during evaluation, the constant is $\alpha_2$.

We differentiate the strength into two parameters because meta-gradients are less stable during the iterative process. This instability can accumulate across layers and episodes, so we aim to apply a lower strength during extraction, if necessary, to mitigate this issue.

**Guide to tuning the hyperparameters.** We recommend a higher value of $k$ for tasks in which the LM demonstrates greater proficiency. Exploring the range of $k \in \{1, 2, 3, 4\}$ is both straightforward and effective, as demonstrated in our experiments, assuming sufficient time is available.

Concerning batch size, we have demonstrated that it should neither be too large nor too small. We recommend starting with $b = 5$ or $b = 10$. Methods for tuning typical batch sizes may also be considered.

Regarding the strength parameters $\alpha_1$ and $\alpha_2$, we performed a comprehensive grid search within the range $[0.1, 0.9]$. Future research is encouraged to employ more sophisticated search strategies, as these parameters often cluster in a low-performance consecutive area (see Figure 3), which can be pruned if properly identified.

