# OpenReview forum: "Iterative Vectors: Boost In-Context Learning within Activations"
_ICLR.cc/2025/Conference — Submitted to ICLR 2025_

### Official Review · Reviewer_mGxX · 2024-10-26

**Soundness:** 4
**Presentation:** 3
**Contribution:** 4
**Rating:** 8
**Confidence:** 3

**Summary:**

The paper discusses the use of Iterative Vectors (IVs) to enhance In-Context Learning (ICL) in language models. IVs are shown to improve performance in various tasks, particularly in few-shot learning scenarios. The paper compares IVs with other vector methods (TV, FV) and demonstrates its effectiveness in reducing inference time and scaling with the number of demonstration shots. The results suggest that IVs offer advantages in improving model performance and can potentially be applied to more advanced applications.

**Strengths:**

The paper shows rigorous process of the research methodology, experimental design, and result analysis. The study rigorously evaluates IVs across multiple models and diverse tasks, providing a comprehensive assessment of their performance.

The clarity of the paper is evident in the detailed explanation of IV generation, their iterative reintroduction into the model, and their impact on ICL performance. The use of clear and concise language, along with illustrative examples, enhances the readability and understanding of the paper. Additionally, the comparison with other methods and the discussion of practical applicability contribute to the overall clarity of the research.

By demonstrating the effectiveness of IVs in improving ICL performance, the paper addresses a critical aspect of language models related to few-shot learning and adaptation to real-world datasets. The practical implications of IVs in enhancing model performance, reducing inference time, and scaling with the number of demonstration shots underscore their significance in real-world applications.

**Weaknesses:**

Clarity could be improved in certain figures and tables.

Table 1: Include brief reminders of the terms FV and TV directly within the table, as they are first defined in Section 2.1, which is distant from their use here. Providing concise definitions and a comparison with IV will improve comprehension.

Figure 3: To improve readability, round the score to one decimal place, which should make the figure appear less dense and easier for readers to interpret at a glance.

Table 4: Converting the results to a plot could visually emphasize the effectiveness of IVs in extracting and utilizing a greater number of examples. This format may better showcase the comparative strengths of IVs.

To make the experiment more comprehensive, consider expanding the experiment in Inference Time Experiment to evaluate FV and TV on the emoji dataset. Assessing their respective performance and inference times would offer a clearer picture of each method's efficiency in various contexts.

**Questions:**

Could you provide further details on the emoji dataset? Additional examples and explanations might help clarify its structure and significance.

The paper primarily evaluates models with relatively small parameters (6B, 7B, and 13B). Would extending experiments to include larger models, such as 70B, offer additional insights or strengthen the findings?

---

> ### Author Response · Authors · 2024-11-25
> **Response to Official Review by Reviewer mGxX**
>
> We sincerely thank the reviewer for the positive feedback and high rating of our paper. We are grateful for their thoughtful review and are pleased to know that they find our work valuable. The suggestions provided are greatly appreciated.
>
> **W1 (Table 1)**: We will added a comparison section in Appendix D.
>
> **W2 (Figure 3)**: We will revise the figure as suggested.
>
> **W3 (Table 4)**: We tested using a plot instead of a table; however, we found it to be a less efficient use of space, which is a concern raised by Reviewer WqMZ. Consequently, we have decided to retain the table. We appreciate the reviewer's suggestion nonetheless.
>
> **W4 (inference time)**: We will add FV and TV to Table 2. Additionally, we have also added the extraction time.
>
> **Q1 (emoji dataset)**: The *emoji* task involves predicting the most likely emoji for a given tweet. This task only considers tweets containing a single emoji, regardless of its position, which serves as the classification label. The dataset for this task is readily available for download and viewing on [HuggingFace](https://huggingface.co/datasets/cardiffnlp/tweet_eval/viewer/emoji/), identified by the tag `tweet_eval/emoji`.
>
> **Q2 (larger models)**: We will incorporate experiments using the `llama-2-70b` model. However, due to our computational budget constraints, we are unable to complete all tasks with a model of this size. Therefore, we have chosen to conduct a multi-shot experiment similar to Table 3 to more effectively demonstrate the efficacy of IV.

---

> > ### Comment · Reviewer_mGxX · 2024-11-25
> >
> > I would like to thank authors for making extra effort to address my comments. After reading their response to my reviews and other reviewers I am willing to keep my score as is (8). Thanks for the great work.

---

### Official Review · Reviewer_WqMZ · 2024-10-30

**Soundness:** 2
**Presentation:** 2
**Contribution:** 2
**Rating:** 5
**Confidence:** 3

**Summary:**

This paper introduces Iterative Vectors (IVs) for enhancing in-context learning (ICL) by exploring the potential of activation space without the need of backpropagating and with minimal computation overhead.

IVs are generated by extracting the difference of attention activations from queries with and without preceding examples during the inference process, with the goal of capturing the insights the model learns from the input examples. These IVs are then iteratively reintroduced into the model, facilitating the formation of more stable and effective vectors while continuously incorporating information from subsequent examples.

The authors evaluate IVs across three models and different classification tasks, reporting improvements over previous  vector-based ICL methods Function Vectors and Task Vectors. They also demonstrate how IVs reduce inference time and scale with the number of demonstrations and extraction episodes. An ablation study is provided to examine the impact of hyperparameters on ICL stability and performance.

**Strengths:**

This work has several notable strengths regarding its topic and results:
- S1 -  The paper addresses an interesting topic of in-context learning (ICL), where authors are specifically focusing on task vectors that enable fast and robust ICL performance. The method Iterative Vectors (IV) contributes to the ongoing research in this direction by further supporting the use of task vectors instead of standard ICL.
- S2 - The authors demonstrate that their method, Iterative Vectors (IV), achieves strong performance over 3 different models and 13 tasks, outperforming previous vector-related approaches without requiring additional memory or backpropagation.

**Weaknesses:**

The paper has several areas that could benefit from improvement in terms of clarity, novelty, evaluation, and formatting:
- **W1** - The introduction lacks a complete motivation for the use of task vectors, making it challenging to understand their necessity in in-context learning.
The writing appears unpolished and the paper should be improved structural-wise. For example, Section 3 discusses the theory of transformers as meta-optimizers, but its connection to the main contributions and approach of IVs is unclear. I would suggest either integrating this better with more related works on this topic, or focusing more directly on the proposed method.
Next, the paper lacks clarity in some parts. For example, the method description could be clearer, especially regarding the hyperparameters, which are mentioned, but not explained properly. Figure 2 also seems to be redundant.
Moreover, there are formatting issues, such as excessive empty space around Figure 3 and within the related work section.  Reformatting the text and refining the introduction would improve the overall presentation and clarity.
- **W2** - The related work section misses important studies on in-context learning, including research on demonstration sensitivity, brittleness, and pretraining dynamics [1, 2, 3] - areas relevant to task vectors. Additionally, it does not reference recent work on task vectors in visual models [4], multi-modal models [5], and VQA [6]. Including these would provide a more comprehensive overview of the field and the paper would be more complete.
- **W3** - Figure 3 is difficult to interpret due to small font sizes and a lot of details and information. I suggest the authors simplify this figure or move it to the appendix while summarizing key findings in the main text to enhance comprehension and clarity.
- **W4** - The proposed method appears to be an extension of existing task-vector based approaches, with slight modifications such as aggregation and different prompt designs applied to new tasks. Further, the method relies on several hyperparameters that require tuning, which might not be practical in few-shot in-context learning situations where minimal parameter adjustment is preferred. Have the authors conducted some additional analysis of the feature space or the underlying mechanisms that could strengthen the paper’s contributions?
- **W5** - The experiments are limited to classification tasks and do not extend to more complex tasks that reflect real-world scenarios or involve multi-modal models. Moreover, the paper does not include evaluations on newer large language models like Llama 3. Have the authors tested different model sizes? If so, how much does the performance differ in such scenarios?
- **W6** - The finding that performance improves with more iterations and demonstrations is somewhat expected in small-data scenarios and few-shot learning. Prior research on multi-modal [5] and visual task vectors [4] has reported similar results. Have the authors conducted more analysis in this direction that can bring new insights for this direction?

[1] https://proceedings.mlr.press/v139/zhao21c.html

[2] https://arxiv.org/abs/2202.12837

[3] https://arxiv.org/abs/2205.05055

[4] https://arxiv.org/abs/2404.05729

[5] https://arxiv.org/abs/2406.15334

[6] https://arxiv.org/abs/2406.13185

**Questions:**

While the paper introduces an interesting approach with Iterative Vectors (IVs) that shows performance improvements over existing methods for vector-based in-context learning methods, there are several areas that require improvement. Strengthening the motivation, refining the method description, expanding the related work to include recent studies, and providing more comprehensive evaluations on a variety of tasks and models would significantly improve the paper. With these revisions, the work could become a strong contribution to the community.

My further questions to the authors are:
- Could you evaluate your method on more challenging tasks that are closer to real-world applications to demonstrate its broader applicability?
- Could you evaluate your method on multi-modal models?
- How does your method perform on newer language models, and is there a significant difference in performance between smaller and larger models?
- Have you analyzed what more iterations and demonstration do to the feature space of the task vectors?
- Have you observed compositionality with Iterative Task Vectors?

---

> ### Author Response · Authors · 2024-11-25
> **Response to Official Review by Reviewer WqMZ**
>
> We're grateful for the effort the reviewer invested in identifying the areas for improvement. We have reorganized our response to the concerns according to their logical structure and will address the questions as follows.
>
> **W1**:
>
> > Section 3 discusses the theory of transformers as meta-optimizers, but its connection to the main contributions and approach of IVs is unclear.
>
> We regret that the reviewer appears to have misunderstood how our method is derived from the interconnected theoretical foundations. We believe this is a crucial aspect of the paper for comprehending the motivation behind our method. As noted by Reviewer sVxj, "the framing of iterative vectors as a way to capture the meta-gradients that occur during ICL is well-done and well-motivated (Section 3.1)."
>
> > Figure 2 also seems to be redundant
>
> As discussed in our response to weakness 1 from Reviewer QraR, Figure 1 serves as a general illustration of how activation vectors function, while Figure 2 specifically demonstrates our proposed method. We consider Figure 2 an essential part of our paper.
>
> > Moreover, there are formatting issues, [...] would improve the overall presentation and clarity.
>
> We appreciate the reviewer's diligent efforts in identifying how adjusting the empty spaces can enhance the clarity of our paper. We will make the necessary revisions.
>
> **W2** & **Q2**: Our research aims to introduce a new direction rather than exploring prompt instability, as examined in previous studies [1-3]. We have, in fact, already cited two of the three papers referenced by the reviewer.
>
> Regarding the works on multi-modal and visual models [4-6], while we appreciate the suggestion, our paper is entirely focused on NLP. The background, motivation, theories, evaluation protocol, baselines, and experiments are all firmly rooted in the domain of NLP. Introducing content related to CV seems incongruous with the rest of our paper and might confuse the readers.
>
> **W3**: We recognize that Figure 3 can be challenging to interpret and will simplify the figure.
>
> **W4**: Upon closer examination, it becomes evident that FV, TV, and our IV differ significantly in the construction of their vectors and the underlying rationale. This distinction goes beyond simply adding aggregation or tweaking prompt design to a previous method. A more detailed comparison of these methods will be provided in Appendix D to elucidate this further.
>
> **W5** & **Q1** & **Q3**: Evaluating a single modality does not imply that the tasks are not based on real-world scenarios. We have incorporated Llama-3.1-8b into our main experiment, as shown in Table 1, where it has demonstrated similarly promising results. Furthermore, our newly added experiments with Llama-2-70b in Table 8 corroborate our findings. This demonstrates that our method is effective across a range of models and model sizes, including one of the more recent models.
>
> **W6**: As stated in the Introduction, increasing the number of in-context examples does not necessarily improve performance. For further details, please refer to our response to question 2 from Reviewer sVxj.
>
> **Q4** & **Q5**: No, but there are valid reasons for this decision.
>
> As foundational work, previous research on FV and TV has naturally focused on identifying the properties of activation vectors, such as feature space and compositionality. In contrast, our approach prioritizes successful adaptation and application to real-world ICL tasks. Consequently, we have deliberately set aside the exploration of feature space to more thoroughly demonstrate the efficacy of our method within the constraints of limited pages.
>
> Feature space investigation is not strictly essential for the application of a method, nor does it always provide useful insights (see the "English-French" row in Table 5 from the FV paper).
>
> Furthermore, when evaluating real-world datasets, the process of composition is not well-defined, as we are no longer dealing with simple synthetic tasks involving easily deconstructed and recombined word pairs.

---

> ### Comment · Reviewer_WqMZ · 2024-11-28
>
> I thank the authors for the detailed responses and additional clarifications. The revised paper and the method are now clearer.
>
> After reading all the rebuttals as well as the revised paper,I have decided to raise my score to 5.

---

### Official Review · Reviewer_sVxj · 2024-11-03

**Soundness:** 3
**Presentation:** 1
**Contribution:** 2
**Rating:** 6
**Confidence:** 4

**Summary:**

The authors introduce iterative vectors (IV), a method designed to enhance the in-context learning (ICL) performance of language models during inference. Their main comparison is with the standard ICL approach, where $k$ examples are included in the prompt, leading to increased performance and computation costs as $k$ increases. In contrast, IV captures the meta-gradients made by in-context examples in a condensed format, which can be applied during inference. The computational cost of these updates is incurred mostly during the creation of IV and is amortized over multiple inference rounds. The authors conduct several experiments to showcase the advantages of using IV in various ICL tasks.

**Strengths:**

1. The framing of iterative vectors as a way to capture the meta-gradients that occur during ICL is well-done and well-motivated (Section 3.1).
2. This paper addresses an important problem: efficiently boosting LM performance without relying on longer prompts.

**Weaknesses:**

1. Presentation: Section 3.3 is difficult to follow. In particular, lines 298-317 are meant to describe how the iterative component of IV works, but I could not understand it even after multiple readings. As currently written, this section heavily relies on formulas to communicate how IV works. There is nothing necessarily wrong with including the formulas, but the surrounding text lacks clarity. Consider adding a new figure, pseudocode, or examples.
2. Missing experiment: Line 36 states, “This finding is also corroborated by our experiments, wherein adding more in-context examples does not always result in improvements. Instead, it introduces uncertainty, which compromises LMs’ reliability and usability. I could be mistaken, but do not see any experiments that support this claim.
3. Missing zero-shot experiments: Given the framing in Section 3.1, it seems natural to include zero-shot experiments. I recommend that the authors either include them or provide an explanation in the paper justifying why these experiments were left out. The authors allude to a reason in line 482, but I did not find it compelling.

**Questions:**

1. Please clarify how the iterative component of IV works.
2. Which experiment supports the claim in line 36?
3. Where is C in formula 17 defined?
4. Please clarify why only the one-shot setting was considered for Table 1. I do not follow the reasoning in lines 350-351.
5. How are iterative vectors different from task and function vectors? A short section comparing their differences will help me better understand the the technical novelty of iterative vectors.

---

> ### Author Response · Authors · 2024-11-25
> **Response to Official Review by Reviewer sVxj**
>
> We are impressed by the reviewer's in-depth review and their thorough examination of the details of our method. We are deeply grateful for their commendation on the effective and well-motivated capture of the meta-gradients, as well as their acknowledgment of the importance of the problem we address. We will strive to provide clear answers to their direct and concise questions.
>
> **W1** & **Q1**: We recognize that formulas can be challenging to understand, and we apologize for any lack of clarity. To assist with this, we will provide an appendix that includes pseudocode to elucidate our method, including the iterative procedure.
>
> **W2** & **Q2**: We apologize for the omission of evidence supporting this claim. We considered it a relatively common instability for standard ICL and did not specifically address it, as it frequently occurred in our experiments. However, this is suggested in Table 2 (on the `emoji` dataset), where the 3-shot clean result is inferior to the 2-shot one.
>
> Here, we include several other experiments that demonstrated this phenomenon. The results are evaluated based on 10,000-episode clean runs on `llama-2-7b`.
>
> | Shot | emoji | stance_climate | emotion | trec |
> |---|:---:|:---:|:---:|:---:|
> | 1 | 9.13 | 28.60 | 54.45 | 74.93 |
> | 2 | 12.90 | 31.26 | 62.13 | 81.18 |
> | 3 | 12.64 | **31.28** | 64.94 | **82.35** |
> | 4 | 13.11 | 29.55 | **66.42** | 81.81 |
> | 5 | **13.34** | 31.22 | 65.81 | 81.56 |
>
> **Q3**: We intended to let $\mathcal{C}_j$ represent the set of indices of the pairs $(x, y)$ that belong to class $j$ with the accompanying text description. We apologize for the lack of precision and have reformulated it to adhere to the convention recommended by the conference. We now redefine it formally as follows:
> $$\mathbb{C}_j = \\{ i \mid \operatorname{Class}(x_i)=j \\}$$
> where $\operatorname{Class}(\cdot)\in \\{1,2,...,n,q\\}$ was previously specified below Formula 24. We will update the paper to reflect changes made to the formulas.
>
> **W3** & **Q4**: Lines 350-351 indicate that, rather than offering the model imbalanced demonstration examples—as is common in studies examining the instability of ICL—we present one example from each class to ensure simplicity and completeness. However, this does not address the reviewer's concern regarding zero-shot experiments. We will expand upon the rationale mentioned in line 482, providing a more detailed explanation of why zero-shot experiments are not included, in Appendix E.
>
> **Q5**: We appreciate the reviewer's suggestion and will include a detailed comparison in Appendix D.

---

> ### Author Response · Authors · 2024-11-30
>
> We would like to follow up on our recent rebuttal comments. Your feedback is invaluable to us, and we are eager to address any remaining concerns. Thank you for your time and consideration.

---

> > ### Comment · Reviewer_sVxj · 2024-12-02
> >
> > Thank you for the thorough response and revisions. I believe the paper is strengthened, and I have raised my score.

---

### Official Review · Reviewer_QraR · 2024-11-04

**Soundness:** 2
**Presentation:** 1
**Contribution:** 2
**Rating:** 5
**Confidence:** 3

**Summary:**

The paper introduces Iterative Vectors (IVs), a method designed to enhance in-context learning (ICL) within large language models (LLMs) by leveraging activation steering in the activation space rather than in the discrete prompt space. The authors propose extracting activation vectors from the difference in model activations for queries with and without prior context examples, and iteratively reapplying these vectors during inference to improve ICL performance. This approach is evaluated on multiple models and diverse classification tasks, demonstrating performance gains over traditional ICL and other activation vector methods.

**Strengths:**

1. The concept behind IVs is simple and straightforward, and the approach of leveraging activation vectors in the model’s activation space intuitively makes sense.

**Weaknesses:**

1. The paper suffers from poor clarity in both writing and figures. For instance, the contributions listed in the Introduction are not clearly differentiated, making it difficult to identify the unique impact of the proposed approach. Additionally, Figure 1 is difficult to interpret and does not clearly explain the IV process. Both the writing and presentation need substantial refinement for the paper to be clear and accessible.
2. The approach is evaluated exclusively on classification tasks, which restricts its generalizability. No experiments are conducted for tasks involving multi-token responses or other types of reasoning, limiting the broader applicability of the findings.
3. The theoretical section does not provide sufficient insight into why IVs improve ICL performance. Although the paper claims that meta-gradients from in-context examples may not fully capture the task, it lacks an explanation of how IVs solve this limitation.
4. More thorough justification and explanation would be necessary for the approach to be convincing.

**Questions:**

1. Could the authors add a written-out algorithm to clarify the procedure for extracting and applying IVs? This would improve reproducibility and understanding of the method.
2. In the theory section, what specific mechanisms in IVs address the purported limitations of ICL in the discrete prompt space? A more detailed explanation here would add to the paper’s depth.
3. Given the observed performance drops with a low number of extraction episodes, are there any strategies the authors could suggest to improve stability in such contexts?
4. The paper notes that extraction and inference strength values play a critical role in performance. Could the authors explain the role of each hyperparameter in a clearer manner? Detailed guidance on tuning these values would improve the usability of the proposed method.

---

> ### Author Response · Authors · 2024-11-25
> **Response to Official Review by Reviewer QraR (part 1/2)**
>
> We appreciate the reviewer's acknowledgment of the simplicity and intuitiveness of our proposed method. We will now address the concerns raised.
>
> **W1**: We apologize for the confusion. Regarding the contributions, we highlighted three key points: our adaptation of previous works to this new direction, our unique invention—a novel activation vector method—and the state-of-the-art performance of our method. Additionally, we emphasized that we are the first to apply activation vectors to diverse real-world ICL tasks.
>
> Figure 1 serves as a basic, general illustration of how various activation vectors operate within the activation space, rather than providing a detailed explanation of our IV. In contrast, Figure 2 is specifically designed to demonstrate the workings of IV. We will revise the content and caption of Figure 1 to clarify this distinction.
>
> **W2**: We appreciate the reviewer's acknowledgment of our statements in the Limitations section. Although our focus is primarily on classification tasks, it is important to note that previous studies often concentrate on synthetic classification tasks or a single type of generative task, such as sentiment transfer or truthfulness elicitation. In contrast, we "rigorously evaluate IVs across multiple models and diverse tasks, providing a comprehensive assessment of their performance", as noted by Reviewer mGxX.
>
> **W3 & Q2**: We regret that the reviewer did not find our derivation—demonstrating how IV more effectively captures the meta-gradients and addresses the limitations of ICL—to be sufficiently convincing. To resolve this, we provide further explanations:
>
> 1. Meta-gradients derived from limited in-context examples may not fully capture the task, instead, they can be adversely influenced by **outlier examples**. To address this issue, IV extracts gradients from multiple episodes and averages them. Achieving this with a large number of examples using an ICL prompt would be costly, if not unfeasible.
> 2. Meta-gradients derived from limited in-context examples may not **scale appropriately** with the original parameters. The gradients might be either too weak to effectively influence the original parameters or too strong, potentially destabilizing them. To address this issue, we introduce the strength hyperparameter $\alpha$. For further insights on this design, please refer to our response to **Q4** below.
> 3. The extraction is conducted in an **iterative** manner, as analyzed in Section 4.4 and illustrated in Figure 3 (where $b$ changes from 0 to 1), resulting in a significant performance enhancement. The vectors are incorporated into the activations during the inference process, achieving a boost that is unattainable within the constraints of the discrete prompt space.
> 4. In addition to averaging the extracted vectors, the iterative updates performed by IV are also **batched**. This standard approach stabilizes the gradients by averaging them over multiple episodes before applying them, thereby reducing the variance in gradient estimates. Consequently, more reliable and consistent updates can be achieved with an appropriate selection of $b$ during extraction.
>
> We hope these explanations will assist the reviewer in recognizing the capture as well-executed, as noted by Reviewer sVxj.
>
> **W4**: In addition to the explanations provided above, we will conduct experiments using a newer model, `llama-3.1-8b`, as well as a larger model, `llama-2-70b`, to further substantiate our approach.

---

> ### Author Response · Authors · 2024-11-25
> **Response to Official Review by Reviewer QraR (part 2/2)**
>
> **Q1**: As the reviewer suggested, we agree that including a detailed algorithm is beneficial and will add pseudocode in Appendix A. Furthermore, we intend to make our code publicly available in the future to support ongoing research.
>
> (**Q2** is addressed in conjuction with **W3** above.)
>
> **Q3**: We appreciate the reviewer's keen observation regarding the initial performance drops in Table 4. The solution might be more straightforward than anticipated: using standard ICL instead.
>
> As demonstrated in Table 4, this drop appears when only two episodes (equivalent to 2-shot) are available for inference, the IV performance is inferior to the standard 1-shot ICL. In scenarios with such a limited number of examples, complex strategies may not even be necessary; a straightforward selection of examples can suffice to achieve standard ICL performance, although outcomes may vary.
>
> IV aims to incorporate a moderate number of examples to address the instability and ineffectiveness that arises when standard ICL is performed on these examples. This means that examples cannot be casually selected nor incorporated into a single prompt, while still maintaining minimal temporal overhead. As evident in Table 4, when more than two episodes are available, IV surpasses 1-shot ICL, thereby fulfilling its intended purpose.
>
> It is important to recognize that the results presented in Table 4 were obtained using a fixed set of hyperparameters, which may not be optimal for each episode amount. It is advisable to search for improved hyperparameters if IV were to be employed.
>
> As an additional note, data augmentation techniques could enhance stability in these contexts. By generating additional training examples through transformations or modifications of existing data, these techniques may provide further benefits. However, this topic is not confined to IV and extends beyond the scope of our paper.
>
> **Q4**: We appreciate the suggestion. We will provide a detailed explanation of the hyperparameters, accompanied by a guide to tuning the hyperparameters, in Appendix F.

---

> > ### Comment · Reviewer_QraR · 2024-11-26
> > **Official Comment by Reviewer QraR**
> >
> > I thank the authors for their responses and added changes to the revised version of the paper. I will increase my rating from 3 to 5.

---

### Official Review · Reviewer_P6ha · 2024-11-04

**Soundness:** 3
**Presentation:** 3
**Contribution:** 2
**Rating:** 6
**Confidence:** 4

**Summary:**

This paper proposes an interesting idea for improving the efficiency of in-context learning (ICL). Building on former observations like Function and Task Vectors, this work invented an iterative procedure that mirrors gradient-based training. The authors compare this method with the two predecessors and standard ICL on multiple models and datasets, showing accuracy improvement. They also study the effect of the number of shots and extraction episodes on this method.

**Strengths:**

* This paper has performed experiments in a principled way, using multiple datasets and base models, and conducted hyperparameter searches for the proposed method and previous method alike.

* This paper is well-written. The motivation and most of the details of the method are well explained.

**Weaknesses:**

* I'm unsure how to interpret some experiment results; more details in the questions.

* The functionality of this method (IV) overlaps with PEFT, but it lacks comparison with any of the PEFT methods. Both IV and PEFT take a small number of training examples, spend some amount of upfront computation (training in the case of PEFT, iterative extraction in the case of IV), produce a small number of additional states (parameters in the case of PEFT, V vectors in the case of IV), and use these states to specify a model more effective on a downstream task. PEFT has already been well adopted and understood. So, if we are to prove this new approach is worth exploring, we need to show that it has unique advantages or better outcomes. (e.g., measuring or analyzing the memory requirement could be helpful.)

**Questions:**

* When we extract the vector iteratively, are the examples encountered in the new iteration the same as the first iteration or resampled every time?

* The FT paper also performed experiments on agnews and sst5, with GPT 1.3B and GPT2.7B, which should be weaker than the models in this paper. But they had sst5 = 39.5, agnews = 65.3 for GPT1.3B, and sst5 = 39.1, agnews = 65.7 for GPT2.7B, significantly higher than the numbers in Table 1. I would guess there are some differences in your experiment setting. I wonder if you could test your method with their setting and report the numbers.

* If we want to test the inference time reduction, why do we need to introduce a brand new dataset? I would guess it will be more informative to use established datasets.

---

> ### Author Response · Authors · 2024-11-14
>
> Thank you for your insightful review! Before we address your concerns, could you please specify which "FT paper" you are referring to (in the second question)?

---

> > ### Comment · Reviewer_P6ha · 2024-11-14
> > **Sorry, it should be FV**
> >
> > Sorry, it was a typo. I was referring to the paper that proposed the FV method.

---

> ### Author Response · Authors · 2024-11-25
> **Response to Official Review by Reviewer P6ha**
>
> We appreciate the insightful review provided by Reviewer P6ha and are honored by their assessment that our paper is "well-written", our idea is "interesting", and our experiments are "principled". We address their concerns as follows.
>
> **W2** (Comparison with PEFT): We are pleased to observe that the reviewer has a profound understanding of our method and provides a comprehensive comparison with PEFT methods. However, there is a significant distinction in the "training" aspect that cannot be considered overlapping between the two approaches, as we outline below.
>
> - **IV does not involve training.** PEFT involves additional parameters that are trained through backpropagation, which is costly in terms of both computation and VRAM. Although it is parameter-efficient, it still results in significant memory usage. In contrast, IV only needs vectors to be stored during inference, eliminating the need for training. This disparity makes it challenging to fairly compare IV with PEFT methods, leading us to choose FV and TV as baselines. Nevertheless, we conducted empirical tests on `llama-2-7b` with the `ag_news` dataset and found the following results:
> 	- A clean run requires 15.2 GiB of VRAM.
> 	- Training a dummy LoRA module with only 0.06% of parameters uses 19 GiB of VRAM after the first episode and exceeds 30 GiB as the prompt sequence lengthens with more verbose episodes. While checkpointing may help reduce VRAM usage, it also results in increased training time.
> 	- IV requires 15.7 GiB of VRAM. The minor surplus is partly due to our retention of vectors from all episodes to facilitate coding and research efforts. This is unnecessary for deployment, as only a vector average is required.
> - **Therefore, IV can be applied in an online manner.** The absence of a training phase allows IV to be used immediately. Upon receiving a batch of examples for inference, IV can iteratively extract and refine vectors without interrupting the inference process, virtually dedicating all computational resources to solving the task without spending them on an upfront training phrase.
> 	- Although determining hyperparameters might be a concern, this issue also exists in PEFT methods, which introduce their own hyperparameters, not to mention those required by the training process—something IV does not need to address.
> - **Moreover, IV requires significantly fewer additional states.** The Iterative Vectors, each corresponding to the model's hidden dimension $D$, are extracted for each of the $L$ layers, resulting in a total additional state size of only $L \times D$.
> 	- For PEFT methods, such as LoRA, the additional state size may be reduced to around 0.1% of the model size.
> 	- For IV, the vectors amount to less than 0.002% of the model size, which is essentially negligible.
>
> In light of the points discussed, we consider the study of activation vectors to be an even more lightweight approach than PEFT that merits separate exploration.
>
> **Q1**: They are resampled each time to continuously incorporate information from more examples.
>
> **Q2**: Regrettably, we were unable to locate any references to "GPT 1.3B", "GPT 2.7B", or the provided numbers in the FV paper available on either [ArXiv](https://arxiv.org/abs/2310.15213v2) or [OpenReview](https://openreview.net/forum?id=AwyxtyMwaG), despite thoroughly reviewing the main text and conducting multiple searches through all supplementary materials. Upon reviewing the code, we confirmed that the [GitHub file](https://github.com/ericwtodd/function_vectors/blob/874d6e93c099d71fe4a2d76551fab233e60062c2/src/utils/model_utils.py) for FV responsible for loading models have not implemented these two models. Additionally, Table 9 in the FV paper, which provides a summary of all the datasets used, does not include SST-5; instead, it mentions SST-2.
>
> Still, we have invested considerable effort into optimizing the hyperparameters of FV, and we have now achieved more satisfactory results. However, this improvement comes at a substantial cost in terms of time, as its design leads to extremely slow extraction processes. Please refer to the updated results in the paper and the limitations of FV in incorporating additional examples in the main experiment, detailed in Section 4. An example of the extraction time required by FV will be included in Table 2, with a discussion of its design provided in Appendix D.
>
> **Q3**: We apologize for the misunderstanding. The *emoji* dataset, designated by the HuggingFace tag `tweet_eval/emoji`, is an established dataset that accompanies the various other datasets we use from the TweetEval framework. We will revise the wording in Section 4.1 where it is first mentioned and ensure it is appropriately included in the table of all datasets.

---

> ### Author Response · Authors · 2024-11-30
>
> We have revised our paper in accordance with the reviewers' feedback and would greatly appreciate your response. Your insights are essential for the enhancement of our work, and we welcome any additional comments or approval.

---

> > ### Comment · Reviewer_P6ha · 2024-12-02
> >
> > Thanks for the explanation. Regarding the results, this is becoming a horror story. I remember I copied those numbers from a table on one paper, but I can't find them anymore. Maybe it's a different and irrelevant paper I opened up when reviewing. Since the results are no longer a concern, I have adjusted the soundness and overall rating.

---

### Author Response · Authors · 2024-11-25
**Summary for Responses**

We have meticulously addressed all the feedback provided by the reviewers and have revised our paper accordingly. We kindly invite the reviewers to first read our summary of the changes made to the paper, followed by our responses to their specific comments. Afterward, we encourage the reviewers to review the revised paper, along with our responses to the other reviews.

The changes we have implemented are as follows:

- **Added Llama-3.1-8b and updated Table 1** (Reviewer P6ha, QraR and WqMZ): Since Llama-3.1 is relatively new, we needed to update our code dependencies to accommodate it. This caused our previous code to break, necessitating updates and a rerun of the main experiment to ensure reproducibility. Additionally, we dedicated considerable effort to enhancing the results on FV by extensively optimizing its hyperparameters, leading to more satisfactory outcomes. For further details, please refer to our response to question 2 from Reviewer P6ha, as well as the initial portion of Section 4 in the revised paper.
- **Added two other metrics** for the main experiment in Appendix C.
- **Added Llama-2-70b in Table 8** (Reviewer QraR, WqMZ, and mGxX): An additional experiment has been conducted using the Llama-2-70b model. Please refer to Appendix C for more details.
- **Updated Table 2** (Reviewer mGxX): We have incorporated FV and TV into Table 2, along with the extraction time for reference.
- **Added pseudocode** (Reviewer QraR and sVxj) in Appendix A. We believe this addition significantly enhances the clarity of our method by providing annotated pseudocode in detail, which complements the formulas presented in Section 3.
- **Added comparison with FV and TV** (Reviewer sVxj and mGxX) in Appendix D.
- **Added clarification regarding zero-shot sequences** (Reviewer sVxj) in Appendix E.
- **Added more explanations on hyperparameters** (Reviewer QraR) in Appendix F.
- Updated Figure 3 to enhance its presentation (Reviewer WqMZ and mGxX).
- Revised the wording in Section 4.4 for improved clarity.
- Added a section for reproducibility statement.
- Other changes in response to the reviewers' comments, as detailed in our replies.

We appreciate the reviewers' insightful feedback, which has significantly enhanced the quality of our manuscript. We believe these changes have strengthened our arguments and improved the overall clarity of the paper. We look forward to further discussions.

---

### Meta-Review · Area_Chair_XcTd · 2024-12-22

**Metareview:**

While the paper presents a new approach leveraging activation vectors for ICL, several issues were raised during the review process. The reviewers noted a lack of clarity. The structure of the paper and the quality of writing also require significant refinement. Furthermore, some reviewers found that the scope of the experiments was too narrowly focused on classification tasks and did not sufficiently address more complex (Rev. QraR). They also suggested referencing additional related works. Although the authors conducted new experiments with larger models and provided clarifications in both the main text and appendices, the paper still needs further polishing in terms of theoretical grounding, broader applicability, and presentation. After considering the authors’ rebuttal and revisions, reviewers have slightly increased their ratings but remained cautious about the overall novelty and impact.

**Additional Comments On Reviewer Discussion:**

See the metareview above.

---

### Decision · Program_Chairs · 2025-01-22

Reject